# Role of Hemidesmosomes in Oral Carcinogenesis: A Systematic Review

**DOI:** 10.3390/cancers15092533

**Published:** 2023-04-28

**Authors:** Jordan Nguyen, Tze Wei Chong, Hafsa Elmi, Jiani Ma, John Madi, Asha Mamgain, Eileen Melendez, Julian Messina, Nikhil Mongia, Sanjana Nambiar, Tsu Jie Ng, Huy Nguyen, Michael McCullough, Federica Canfora, Lorraine A. O’Reilly, Nicola Cirillo, Rita Paolini, Antonio Celentano

**Affiliations:** 1Melbourne Dental School, The University of Melbourne, 720 Swanston Street, Carlton, VIC 3053, Australiam.mccullough@unimelb.edu.au (M.M.); nicola.cirillo@unimelb.edu.au (N.C.); rita.paolini@unimelb.edu.au (R.P.); 2Department of Neuroscience, Reproductive Sciences and Dentistry, University of Naples Federico II, 80131 Naples, Italy; 3The Walter and Eliza Hall Institute of Medical Research, Parkville, VIC 3052, Australia; oreilly@wehi.edu.au; 4Department of Medical Biology, The University of Melbourne, Parkville, VIC 3010, Australia

**Keywords:** hemidesmosome, oral, carcinogenesis, OSCC, integrin, plectin, BPA1, Col17, CD151

## Abstract

**Simple Summary:**

Hemidesmosomes are junctional complexes that contribute to the attachment of epithelial cells to the underlying basement membrane. Importantly, detachment from the basement membrane, migration and invasion through the connective tissues represent early steps in oral carcinogenesis. Therefore, it is possible that these processes involve alterations of hemidesmosomes. The results of our systematic review provide evidence that oral potentially malignant disorders and oral cancer are associated with structural and molecular modifications of hemidesmosomes. We conclude that these cell adhesion structures represent potential candidates for use as biomarkers.

**Abstract:**

Background: Oral cancers have limited diagnostic tools to aid clinical management. Current evidence indicates that alterations in hemidesmosomes, the adhesion complexes primarily involved in epithelial attachment to the basement membrane, are correlated to cancer phenotype for multiple cancers. This systematic review aimed to assess the experimental evidence for hemidesmosomal alterations, specifically in relation to oral potentially malignant disorders and oral squamous cell carcinomas. Methods: We conducted a systemic review to summarise the available literature on hemidesmosomal components and their role in oral pre-cancer and cancer. Relevant studies were retrieved from a comprehensive search of Scopus, Ovid MEDLINE, Ovid Embase and Web of Science. Results: 26 articles met the inclusion criteria, of which 19 were in vitro studies, 4 in vivo studies, 1 in vitro and in vivo study, and 2 in vitro and cohort studies. Among them, 15 studies discussed individual alpha-6 and/or beta-4 subunits, 12 studies discussed the alpha-6 beta-4 heterodimers, 6 studies discussed the entire hemidesmosome complex, 5 studies discussed bullous pemphigoid-180, 3 studies discussed plectin, 3 studies discussed bullous pemphigoid antigen-1 and 1 study discussed tetraspanin. Conclusion: Heterogeneity in cell type, experimental models, and methods were observed. Alterations in hemidesmosomal components were shown to contribute to oral pre-cancer and cancer. We conclude that there is sufficient evidence for hemidesmosomes and their components to be potential biomarkers for evaluating oral carcinogenesis.

## 1. Introduction

Oral cancers are one of the most common malignancies worldwide, with approximately 145,000 deaths and 300,000 new cases reported in 2021 [1], and they come under the umbrelalla of head and neck squamous cell carcinoma (HNSCC). They are defined as malignant neoplasias occurring within the oral cavity or on the lips, with oral squamous cell carcinoma (OSCC) being the most common malignancy, accounting for 92–95% of all oral cancers [2]. Oral potentially malignant disorders (OPMDs) are precursors to oral cancer and constitute a group of benign mucosal disorders associated with an increased risk of oral cancer development. These include leukoplakia, erythroplakia, oral lichen planus, oral lichenoid lesions, and oral submucous fibrosis, among others [3]. Current literature links mucosal alterations to an increased risk of malignant transition in comparison to normal mucosa and those without OPMDs [3], and cancer progression arising from OPMD is thought to be a multi-step process leading to the accumulation of certain genetic mutations [2]. Currently, histopathological features, specifically the presence and degree of epithelial dysplasia, remains the most useful indicator and a clinical diagnostic tool for identifying the risk of cancer progression [4]. Due to the complexity and variability in OPMD presentation, identifying biomarkers of OPMDs capable of predicting the likelihood of progression will aid in early disease diagnosis and improve clinical management [5]. Pertinently, multiple studies have reported alterations in epithelial anchoring junction hemidesmosomes (HDs) in OPMD lesions that may predict progression to malignancy [6]. HDs contain multiprotein structures that facilitate the linkage between basal epithelial cells and the underlying basal lamina. They are also a component of the extracellular matrix and the mechanically stress-resilient keratin intermediate filament (IF) network [7]. Based on their molecular structure, HDs are classified as either Type I or Type II (Figure 1) [8]. Several reports have implicated enzymes, such as matrix metalloproteinases, with the transition to oral carcinoma and invasiveness [9,10]. Alternatively, the disruption of the basement membrane (BM) could result from the inflammatory infiltrate involving neutrophils, fibroblasts, and macrophages within the connective tissue [11,12,13]. However, these possibilities are not yet conclusive and have been suggested to explain the malignant progression and invasion of epithelial cells during oral carcinogenesis and may correspond to limited hemidesmosome expression/function.

This systematic review aims to collate currently available evidence regarding hemidesmosomal alterations during oral carcinogenesis and to address the following research questions: Are alterations to hemidesmosomes or their components correlated with particular stages of oral cancer?Which cellular pathways do hemidesmosomes contribute to during oral carcinogenesis?Can alterations in hemidesmosomal components be used as stage-specific biomarkers for oral cancer progression as well as prognosis?

## 2. Materials and Methods

The 2020 version of PRISMA was used as a guideline for screening and data collection in this systematic review. 

### 2.1. Literature Search Strategy

Electronic literature searches were conducted on 17 June 2022, using Scopus, Ovid MEDLINE, Ovid Embase and Web of Science databases with no publication year restrictions. The search strategy applied was identical across all databases, with the exception of the database-specific syntax. The search term strategy involved keyword searches for all terms pertaining to; (1) hemidesmosomes, (2) oral cavity, (3) stage and development of cancer and (4) the presence of OPMDs [3]. Refer to Appendix A (Table A1) for the full search strategy. 

### 2.2. Selection Criteria

#### 2.2.1. Inclusion Criteria

-Article types: in-vivo and in-vitro studies;-Hemidesmosomes and components (not all adherens junction);-Oral cancer and/or precancerous conditions (human and/or animal);-Published in English;-No restriction on publication date.

#### 2.2.2. Exclusion Criteria

-Article types: Review articles, conference abstracts, letters to the editors, personal communications and opinion articles, and case reports;-Retracted studies;-No full text available.

### 2.3. Quality Assessment

A total of 26 studies were included in the systematic review for quality assessment. The Office of Health Assessment and Translation (OHAT) risk of bias tool, which is designed for both human and animal studies, was used for assessment [14]. The OHAT tool consists of 11 questions and each question has 4 possible answers (definitely low, probably low, probably high and definitely high). The 11 questions were classified into six domains (selection, confounding, performance, attrition/exclusion, detection, and selective reporting). The OHAT questions were adapted to ensure relevance to the included in vitro studies. A total of nine questions were applicable to the quality assessment (Appendix A, Table A2).

### 2.4. Statistical Analysis

All statistical analysis was performed using IBM^®^ SPSS Statistics^®^ (version 27.0). Cohen’s Kappa coefficient was calculated for the two independent reviewers conducting the title and abstract screening [15]. Additionally, Cohen’s Kappa coefficients were then calculated between pairs conducting the full-text screening, and those studies with the complete agreement were included.

## 3. Results

### 3.1. Literature Selection

The process of selection for the inclusion of eligible studies is summarised in a flowchart (Figure 2). The literature search identified 298 papers, which were compiled on Covidence, and a total of 126 duplicates were removed by this software. Two blind reviewers additionally manually extracted undetected duplicates. The remaining 172 unique articles were then assessed for inclusion in the current review.

In accordance with PRISMA protocol, two independent reviewers were selected for the title and abstract screening. Covidence was used for the title screening process. From the 172 articles, 52 studies were selected for inclusion, with a Cohen’s Kappa coefficient of 0.71 (95% Confidence Interval [C.I.]: 0.610–0.824). Conflicts between reviewers were resolved via discussion with an independent third reviewer. Ultimately, a total of 52 papers were unanimously accepted by all the reviewers for full-text screening.

Full text screening was conducted by eight independent reviewers. The eight reviewers were paired, and each group underwent calibration for the screening process. A total of 26 articles were included in this systematic review, with key characteristics (Table A3 and Table A4) and a Cohen’s Kappa coefficient of 1.0 considered for each pair of reviewers. The specific methods used to assess each hemidesmosomal component have also been summarised in Table A3.

### 3.2. Risk of Bias

Two independent reviewers evaluated the risk of assessment for the included studies via the OHAT tool, with a Cohen’s Kappa score of 0.79 (95% [C.I.]: 0.723–0.861). Conflicts were resolved via discussion between the reviewers as summarised in Figure 3. A bias rating of “definitely high” was found for ‘detection’ (4%). A rating of “probably high” was found in categories of ‘performance’, ‘attrition’, ‘detection’, ‘selective’ and ‘other’ (15%, 8%, 6%, 4% and 46%, respectively). A “probably low” risk of bias was evaluated for ‘selection’, ‘performance’, ‘attrition’, ‘detection’, ‘selective’ and ‘other’ (2%, 81%, 92%, 96% and 54%, respectively). Finally, a “definitely low” risk of bias rating was determined for ‘selection’ and ‘performance’ (98% and 4%, respectively). (Refer to Appendix A, Table A2).

### 3.3. Hemidesmosome Complex

A total of six studies, three in vitro and three in vivo, analysed the properties of carcinogenesis in oral tissues in relation to the entire hemidesmosome complex [16,17,18,19,20,21]. These studies, using Electron Microscopy (EM), indicated that in normal oral mucosa sections, hemidesmosomes were observed as highly dense aggregates along the basal cell surface facing the basal lamina. In contrast, severe dysplasia showed a thin and discontinuous basal lamina with few hemidesmosomes adjacent to the basal lamina. During oral carcinogenesis, there was a generalised consensus of a reduction in expression of the hemidesmosome, both fewer in number and appearing smaller with discontinuity of the basal lamina. These findings also correlated with increased invasiveness of epithelial cells, as notably the adherence between the IFs and the BM became more diffuse.

#### 3.3.1. α6β4. Heterodimer

Integrins are transmembrane receptors broadly found within cellular membranes and thereby function to link the cellular cytoskeleton to the extracellular matrix [22]. In the context of HDs, these receptors are composed of an α and a β subunit. The α6 and β4 subunits combine to form α6β4 integrin heterodimers that subsequently act as laminin-332 receptors and link with the basement membrane (BM) [7]. The phosphorylation of the cytoplasmic tail of the β4 integrin subunit enables the α6β4 integrin to be released from the hemidesmosome complex and thus promote invasiveness and metastasis through growth factor signalling cooperation and alterations in gene transcription to further promote carcinogenesis [23].

A total of 15 studies, comprising 12 in vitro, 1 in vivo and 2 in vivo and in vitro studies, examined changes to individual α6 and/or β4 subunits at different stages of oral carcinogenesis [6,24,25,26,27,28,29,30,31,32,33,34,35,36,37]. Expression of α6 in OSCC and oral leukoplakia (OL) lesions was higher compared to normal mucosa, while β4 expression was increased in OSCC lesions with negligible changes in OL lesions [25]. Integrin subunit α6 was also elevated in OSCC lesions in a rat model [30]. Precancerous cellular stages of transformation, such as hyperplasia and dysplasia from human biopsies showed increased expression of α6β4 integrin. However, these subunits decreased during the latter stages of cancer development [29]. Interestingly, β4 upregulation mediated epidermal growth factor receptor (EGFR) signalling, indicating that the β4 subunit may have a role to play in facilitating tumour migration [6]. Furthermore, a number of OSCC studies directly correlated metastasis with upregulated gene expression and elevated protein levels of α6 and β4 subunits within tumour cells [27,31,32].

Pertinently, in humans, increased β4 subunit levels were shown to correlate with higher rates of OSCC mortality [31]. In studies comparing OSCC tumours at various stages of progression, alterations in polarity and reduced numbers of α6 and β4 were observed more in poorly differentiated (PD) tumours compared to moderately differentiated (MD) and well-differentiated (WD) OSCC tumours [24].

A total of 12 studies, comprising 11 in vitro and 1 in vivo, specifically examined alterations in the hemidesmosomal integrin α6β4 heterodimer [24,25,26,27,28,29,30,33,34,35,36,37]. An in-vitro study on human primary OSCC tumours demonstrated increased α6β4 expression in aggressive lesions compared to non-metastatic growths, but reduced expression in PD and recurring tumours [27,34,36]. This observed reduction in the expression of the α6β4 heterodimer in PD tumours is in agreement with the reduced α6 and β4 subunits previously found in vivo [18]. Human in vitro studies of oral lichen planus (OLP) indicated the discontinuous distribution of α6β4 in basal epithelial cells of lesions compared to normal human mucosal cells [26]. Increased levels of α6 and β4 subunits were also found in cells from OLP lesions [33].

A human study of the epithelial–mesenchymal transition (EMT) in oral carcinogenesis demonstrated that the breakdown of the α6β4 complex was associated with cellular alteration, leading to an alternate pairing of subunits with other heterodimers, such as α6β1. This latter complex has been demonstrated to have an affinity with laminin α5 through electrostatic interactions with large surfaces on the β-propeller domain of α6 [35]. Furthermore, transfection of β4 into H376 OSCC cells demonstrated that increasing α6β4 cell surface expression did not improve the proportion of cell adherence to laminin 1 and 5. Furthermore, it also did not increase the capacity of cells to express involucrin, a protein marker of cellular differentiation, often downregulated in OSCC and a constituent of the insoluble cornified cell envelope of stratified squamous epithelia [28].

Overall, these results demonstrate that OMPDs and OSCC are associated with structural modifications of HDs and with molecular alterations of their constituents. Interestingly, our data suggest that a reduction in density and size of hemidesmosomal complexes is paralleled by an increase in the expression of α6 and β4 integrin subunits during oral carcinogenesis, as well as a reduction in α6β4 heterodimer in poorly-differentiated tumours.

#### 3.3.2. Plectin

Plectins and BPAG1, belong to the plakin family of cytoskeletal linker proteins connecting the keratin filaments to the hemidesmosomal junction through α6β4 integrin. Plectin is a protein of >500 kDa, which acts as an integral part of the cytoskeleton network organization, involving the viscoelastic properties of the cytoplasm and the mechanical integrity and resistance of cells. Plectin has also been found to promote the migration and invasion of HNSCC cells through the activation of Erk 1/2. In our review, a total of three studies, two in vitro and one both in vitro and in vivo, examined the role of plectin in oral carcinogenesis. Plectin loss in OSCC-derived cells has been shown to reduce cell migration, invasion and tumourigenicity [38]. Proposed mechanisms responsible for this reduced cell motility and invasiveness include decreased Arp 2/3 protein expression and MMP-9 activity, respectively [38]. Indeed, induced expression of MMP-2 and -9 has been found to be positively associated with HD disruption and wound repair in keratinocytes. Enhanced phosphorylation and upregulation of plectin s4642 along the basal membrane in OLP was also reported, with reticular OLP displaying the strongest expression of cytoplasmic and membranous plectin in basal and suprabasal cells [33], showing an increased interaction with β4 integrin when vimentin (a type III IF) was depleted in OSCC-derived cells [39]. Furthermore, co-immunoprecipitation of plectin and β4 integrin with vimentin was observed in OSCC-derived cells [39]. Overall, these studies suggest that plectin upregulation is associated with tumourigenic progression of OSCC.

#### 3.3.3. BPA1/BPAG1e/BP230/Dystonin

Two in vitro studies examined the role of BPA1 in oral carcinogenesis. Downregulation of BPA1 resulted in reduced tumourigenic potential of OSCC-derived cell lines, through reduced cell migration and invasion, accompanied by changes in actin organisation [38]. In contrast, increased BPA1 gene expression was observed in malignant tumours compared to mildly dysplastic regions and normal squamous mucosa, and in primary tumours compared to non-metastasising tumours [27]. Finally, aberrant loss of basal BPA1 with extended pericellular localisation of BPA1 in the floor of mouth (FoM) OSCC was associated with increased cytoplasmic BPA1 localisation in OLP. In general, these studies indicate that basal BPA1 loss may be a predictive marker for the metastatic potential of invasive OSCC tumour cells [27].

#### 3.3.4. BPAG2/Col17/BP180/COL17A1

Five studies investigated alterations of Col17 in OPMD and OSCC [6,33,40,41,42] and used mostly immunostaining techniques, with the overall findings indicating that OLP cells had higher levels of Col17 in the cytoplasm compared to normal oral mucosa [33]. Additionally, both central and peripheral OSCC tumour cells showed upregulated mRNA expression of Col17, compared to healthy oral mucosa. Similarly, severe dysplastic cells demonstrated upregulated Col17 levels in suprabasal keratinocytes; however, in mild dysplasia, basal cells exhibited downregulation [40]. Col17 also appears to be augmented in OSCC cells, specifically in peripheral tumour nest cells and the cytoplasm of OLP cells, resulting in compromised attachment with IFs that usually stabilise the cell structure and anchorage to the basement membrane [33,40]. Additionally, elevated Col17 expression was associated with greater invasiveness and OSCC severity, possibly due to Col17’s ability to promote transmigration [6,41]. Crucially, increased Col17 positivity may potentially be used as a prognostic factor for OSCC patients, associated with a higher mortality risk at later clinical stages [42].

#### 3.3.5. CD151

CD151 belongs to the tetraspan superfamily proteins associated with other tetraspan molecules and integrins to form large complexes at the cell surface highly concentrated in hemidesmosomes [43] CD151 has been implicated in cell adhesion and motility and the transport of integrins via vesicles [44]. Upon co-transfection with β4, the surface expression of CD151 is enhanced, and the protein becomes recruited into hemidesmosomes. This process only occurs when the α6 subunit is associated with β4, indicating that CD151 is probably bound to this subunit. One in vivo study investigated the expression of CD151 in OLP and found that the location of CD151 was unaltered when healthy controls were compared to OLP mucosa and was expressed along the basal membrane (seven cases) or in the cytoplasm (five cases) [33].

## 4. Discussion

The process of cancer progression and metastatic dissemination requires detachment from the basal membrane. However, there is currently limited evidence regarding the involvement of hemidesmosomal alterations during the progression from oral pre-cancer to cancer. There is also limited evidence regarding a potential correlation between the alterations of the hemidesmosomal subcomponents and overall OSCC prognosis and survival rate. The data presented in this study show that hemidesmosomal alterations do occur in OPMDs, particularly OLP, as well as during the process of oral carcinogenesis, invasion and metastasis. The main trends of expression alteration for each hemidesmosomal components across the oral precancer and cancer disease spectrum are summarised in Table A5.

The link between hemidesmosomal components and oral lichen planus was investigated in three papers [25,26,33]. Expression of α6 and CD151 subcomponents were increased in OLP in comparison to normal mucosal tissues; however, integrin β4 exhibited poor expression [27,33]. Additionally, localisation of α6β4 integrin was diffuse or discontinuous in OLP lesions, suggesting the possibility of α6 polarised expression levels and altered localisation as an early indicator of oral cancer [25,26,33]. α6β4 expression in pre-cancerous conditions, such as hyperplasia and dysplasia, are increased but frequently absent during the later stages of cancer development [29,30]. These studies indicate the possibility of α6β4 and CD151 expression as biomarkers for both pre-cancer and cancer progression.

From our analysis of the current literature, numerous studies have suggested a correlation between the entire hemidesmosomal complex and oral carcinogenesis [16,17,18,19,20,21]. One study identified alterations in the hemidesmosomal proteins β4 and COL17A1 and a change in their localisation from cell membrane to cytoplasm, which was associated with tumour progression within oral tissues. This is suggestive of an underlying intrinsic influence and linkage between the keratin IF network and BM in the advancement of oral cancer [6]. Several studies also reported a numerical loss of HDs, usually coupled with atypical thinning or absence in the BM during tumour progression [16,45]. Supporting this notion, White and colleagues (1980) detected an increase in the size and frequency of lysosomes in basal cells suggesting enzymatic involvement [20]. Overall, our review of the literature pertaining to OSCC is supportive of a role for the entire hemidesmosomal complex in this cancer.

As regards the hemidesmosomal subcomponents, our review has shown that there is supportive evidence for plectin not only as an integral factor binding to β4 integrin and connecting the HD to keratin IFs but also a role in OSSC carcinogenesis. Plectin’s ability to bind to IFs is compromised when plectin S^4642^ is phosphorylated (associated with hemidesmosomes in the epidermis and in hemidesmosome-like structures in cultured keratinocytes), and this coincides with decreased MMP-9 activity [38,46]. Elevated MMP-9 is known to facilitate cancer cell invasion and metastasis as well as potentiate the invasiveness of OSCC [47]. Supporting this theory, plectin loss resulted in reduced cell migration in OSCC-derived cells that also coincided with decreased expression of Arp 2/3 complexes [38]. Furthermore, it was reported that Arp2/3 was expressed more commonly in invasive OSCC than in non-invasive verrucous carcinomas and carcinomas in situ [48]. Moreover, there was a concomitant elevation in actin-related protein (Arp) 2/3 complex levels and colocalisation of β4 integrin and plectin associated with vimentin downregulation in OSCC-derived cells [39]. Importantly, vimentin has been demonstrated as a potential biomarker for OSCC [49]. More conclusively, co-immunoprecipitation studies revealed plectin as a possible linker between β4 integrin and vimentin [39], indicating a role for plectin in OSCC progression.

BPA1/BPAG1e/BPA230/Dystonin is also a component of the hemidesmosome that facilitates linkage to the keratin cytoskeleton [50]. A single study in OSCC-derived cell lines reported that the downregulation of BPA1 decreased cell motility and reduced cell invasion [38]. Similar findings implicate disruption of BPA1 expression through loss of IF attachment to the HD interfering with basal keratinocyte migration [23]. BPA1-knockdown results in changes in actin organisation known to have a role in cell migration regulation [38,51]. Additionally, BPA1-Plectin knockdown cells were less invasive and had significantly reduced MMP9 activity [38]. Elevated MMP activity (MMP2 and MMP9) has been reported to correlate with enhanced invasive potential of OSCC [47]. BPA1 gene expression was also observed at higher levels in malignant tumours and metastasising primary tumours, compared to mildly dysplastic/normal tissues and non-metastasising tumours, respectively [27]. In conclusion, BPA1 appears to be a promising biomarker for assessing OPMD carcinogenic potential. Upregulated synthesis of BPA1 mRNA and α6-integrin are associated with invasive tumours and extended expression of α6β4-integrins. Crucially, BPA1 is associated predominantly with metastasizing rather than non-metastasizing primary tumours.

Two papers mentioned aberrant cellular localisation of BPA1 in OLP [27,33] with increased cytoplasmic BPA1 in basal and parabasal cells. This finding was thought to be due to failed recruitment of BPA1 to HDs [33], leading to the formation of Type II HDs associated with weakened binding to keratin filaments and thus the BM, permitting cell migration [52,53]. In contrast, the loss of basally polarised localisation and enhanced pericellular localisation of BPA1 was demonstrated in an MD FoM carcinoma [27]. It is possible the progression of OMPDs to OSCC involves a shift from basal to pericellular BPA1 localisation, promoting migration and affecting cell growth or substratum adhesion. Indeed, these pathways are involved in compromising the mechanical integrity of the cells and their cancer evolution.

The role α6β4 integrin in the context of OSCC has been investigated in a number of studies to determine if cancer cells are associated with alterations in α6β4 expression and/or polarity [24,27,36,37]. Observed α6β4 mRNA expression was increased in OSCC and upregulated in invasive cancer tissue [27]. These findings suggest that an increase in expression could be associated with a shift from pre-malignant to malignant status and a concomitant decrease with poor differentiation. Furthermore, α6 integrin was proposed as a biomarker to predict the metastatic potential of tumour cells with an absence of α6β4 found in recurring tumours [27,36]. OSCC induced in rat cells through exposure to 4-nitroquinoloine-1-oxide also resulted in oral tumours with the absence of α6β4 compared to normal keratinocytes [37]. Interestingly, it has been postulated that α6β4 integrin released from HDs may promote cancer cell growth, invasion and metastasis [54], possibly through epidermal growth factor receptors, which increase signalling of the PI3K-Akt pathway, the Mitogen-Activated Protein Kinase (MAPK) pathway and the Rho small GTPase pathways [55,56,57,58].

Epithelial–mesenchymal transition is the sequence through which carcinoma cells suppress their epithelial features and acquire mobility and invasive potential. In this regard, Takkunen et al. observed transformation with the removal of the β4 subunit and replacement with β1, thus forming an α6β1 heterodimer [35]. This modification could be used to predict the transition from solid benign tumours to malignancy with increased migrative properties [35]. In another study, Jones et al. attempted to restore the β4 subunit in an OSCC line through transfection with β4. This failed to reestablish stable anchorage and suppress the proliferation of these cells, suggesting that other pathways could also be involved in mediating continued growth and de-differentiation in these cell lines, aside from α6β4 integrin loss [28].

In terms of the Integrin β4 subcomponent, two studies investigated the relationship between the β4 integrin subcomponent and the risk of OSCC mortality [31,39]. OSCC metastasis and higher risk of mortality were associated with a high gene expression ratio of β4/plakoglobin (JUP) [31]. In addition, integrin β4/JUP gene expression ratios were also related to lymph node metastasis (LMN), primary site recurrence, distant metastasis and death from OSCC. This indicates a potential adoption of β4 integrin as a prognostic marker for local invasion and metastasis [31]. Additional studies linked the expression of integrin β4 with nodal metastasis, EGFR signalling and vimentin expression, suggesting migrative capabilities [6,39]. This could possibly explain the heterogeneity of β4 integrin’s role in metastasis. In conclusion, if hemidesmosomes remain intact, they can function as an anchoring unit; however, during cancer progression, the β4 subunit appears to promote cell survival and invasion [55,56,57,58].

Altered expression of Col17 (BP180/Col17A1) during the process of oral carcinogenesis and dysplasia has been reported extensively within the literature. In mild dysplasia, decreased expression was observed with disturbed adhesion of basal keratinocytes within the ECM, while moderate and severe dysplasia, as well as grade II and III patients, displayed increased Col17 without hemidesmosome formation. This indicates Col17 involvement in tumour progression, further evidenced by its upregulation in response to a tumour promoter [40]. The complexity of Col17, composed of a globular cytoplasmic domain (associated with 4 integrins and a large extracellular domain containing multiple collagenous repeats) and associated with the 6 and 1 integrins, is also highlighted and may point towards other potential roles besides cell adhesion [40]. Col17’s complex association with different stages of oral carcinogenesis illustrates its potential as a staging/grading biomarker and prognostic parameter in OSCC. Col17 is upregulated at the invasive front of malignant tumours, and cytoplasmic staining is associated with higher histological grade and invasiveness of OSCC, while unchanged localisation of Col17 may indicate a low probability of LMN [6,40]. Furthermore, the cell adhesion domain of Col17 may stimulate the transmigration of OSCC cells by utilising a molecular interaction with αIIb integrin uncovering its novel chemotactic function in cancer invasion [41]. A hypothetical mechanism for Col17’s correlation with cancer severity is that its extracellular domain is essential for basal lamina function by connecting cytoplasmic structures with the ECM. Disrupting this relationship may influence cell invasion and migration and, therefore, tumourigenesis and progression [42]. Finally, it is postulated that Col17 may serve as a factor for prognosis determination, with positivity suggestive of lower OS and disease-free survival (DFS) scores and a high risk of death in later clinical stages of cancer [42].

## 5. Conclusions

This systematic review evaluated the relevance of HDs and their subcomponents in relation to oral carcinogenesis and overall prognosis. The assessment of correlation and expression of HDs subcomponents may enable clinicians to precisely estimate the extent of local invasion and lymphatic metastasis of OSCC and provide information on the incidence of local recurrence as well as the possibility of a targeted therapy. However, further research needs to be conducted to investigate how alterations in hemidesmosomal components themselves relate to pathways that influence progression to oral cancer. The current body of evidence primarily implicated reduction in mature hemidesmosomal structures and increased expression of its subcomponents with cancer severity and poorer prognosis. These findings suggest that it would be of value in future methodologies that categorise prognosis and cancer severity to include an analysis of HDs and hemidesmosomal subcomponents.

## Figures and Tables

**Figure 1 cancers-15-02533-f001:**
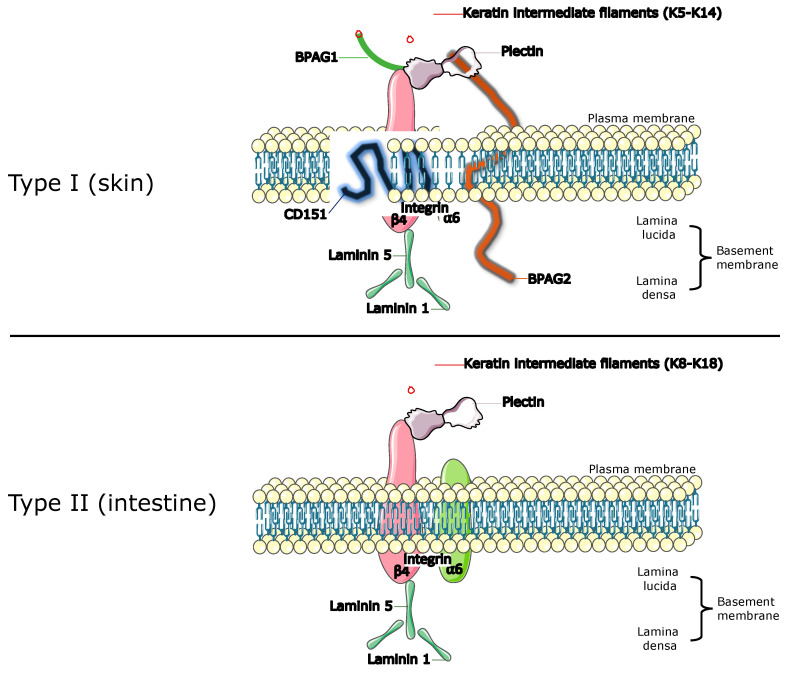
The hemidesmosomal complex exists as two alternative types. Type I consists of Integrin α6β4, Plectin isoform 1a, bullous pemphigoid antigen (BPAG)1 isoform e, BPAG2 and tetraspanin CD151. Type II is formed by Integrin α6β4 and Plectin isoform 1a only. Both are specialised multiprotein junctional complexes that mediate adherence of basal epithelial cells to the underlying basement membrane. (Figure created on Canva).

**Figure 2 cancers-15-02533-f002:**
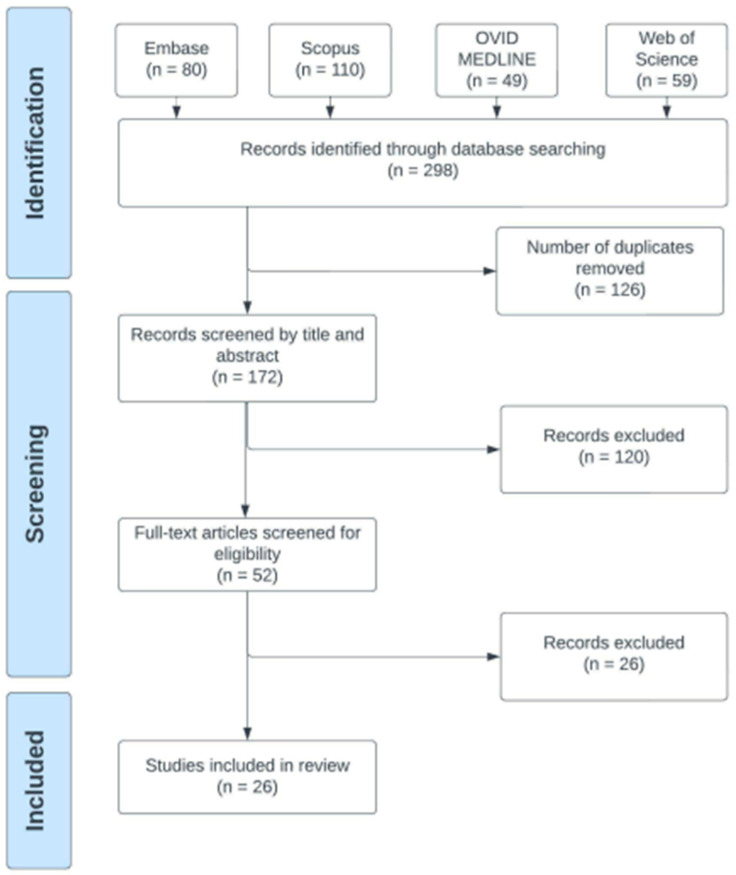
Selection processes for the inclusion of eligible studies.

**Figure 3 cancers-15-02533-f003:**
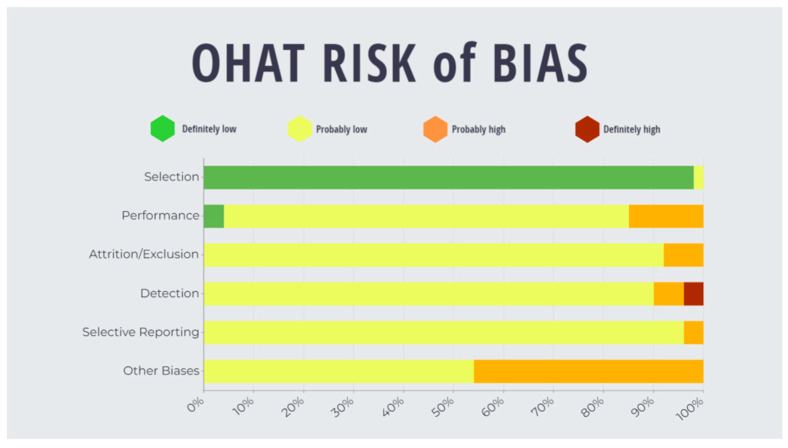
Summary of risk of bias assessment of included studies under OHAT guidelines.

## Data Availability

The data that support the findings of this study are available from the corresponding author upon reasonable request.

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
