# Peer review of "Role of Hemidesmosomes in Oral Carcinogenesis: A Systematic Review"

_cancers, 2023, doi:10.3390/cancers15092533_

Round 1

Reviewer 1 Report

1. In the text where each biomarker is mentioned it would be desirable to add the experimental method used by the author (eg, immunohistochemistry).

2 A summary table listing individual hemidesmosomes (by name) and whether the expression is increased, decreased, or lost and giving the relevant citation would be helpful to the reader. This will also indicate to the reader any agreement/disagreement in the results for each biomarker.

3. Results in lines 178 and 179 (citation 25) appear to be conflicting with the general conclusion. Kindly comment on the discussion.

4. The conclusion can be improved by indicating the translational value of the findings from this systematic review.

Author Response

Reviewer 1

Comment 1: In the text where each biomarker is mentioned it would be desirable to add the experimental method used by the author (eg, immunohistochemistry).

Reply: We thank this reviewer for this comment. However, we would like to point out that an extensive representation of the specific methods used for each biomarker has already been reported by these authors within Table 3. Given the wide range of different methods used for each biomarker, it would be highly redundant and overwhelming for the reader to have this information reiterated in the text. Therefore, we have now recalled this information into the text as a single statement redirecting the reader to the appropriate table (page 5, line 146-147).

Comment 2: A summary table listing individual hemidesmosomes (by name) and whether the expression is increased, decreased, or lost and giving the relevant citation would be helpful to the reader. This will also indicate to the reader any agreement/disagreement in the results for each biomarker.

Reply: Thank you for your valuable comment. As per your suggestion we implemented a summative table reporting effectively the main trends of expression alteration for each hemidesmosomal components across the oral precancer and cancer disease spectrum. This table now reads “Table 5” and is also reported below for your perusal.

Table 5. Summary of the Hemidesmosomal component expression alterations in OPMDs and OSCC across the included studies.

N. of studies with increased expression (Refs)

N. of studies with decreased expression (Refs)

N. of studies with loss of expression (Refs)

OPMDs

OSCC

OPMDs

OSCC

OPMDs

OSCC

Hemidesmosomal complex

0

0

1 (19)

3 (18; 20; 21)

0

2 (16; 17)

α6

1 (25)

2 (25; 30)

0

0

0

1 (29)

β4

1 (33)

1 (25)

1 (25)

1 (32)

0

1 (29)

α6β4

1 (29; 33)

1 (26)

1 (26)

1 (34)

0

1 (36)

Plectin

1 (33)

0

0

0

0

0

BPAG1e

0

2 (26; 33)

0

0

0

0

BPAG2

3 (33; 40;42)

2 (6; 40)

0

0

0

0

CD151

1 (33)

0

0

0

0

Comment 3: Results in lines 178 and 179 (citation 25) appear to be conflicting with the general conclusion. Kindly comment on the discussion.

Reply: Thank you for your valuable comment. Howeber, even if through a fast reading the results appear to be conflicting with the conclusion, this is not the case.

Results, line 178,179: report that there is a reduction of the overall number of hemidesmosomes as a whole structure, with α6 and β4 expression levels increased in OSCC compared to normal, (despite the reduced number of hemidesmosomes).

Conclusions: “The current body of evidence primarily implicated reduction in mature hemidesmosomal structures and increased expression of its subcomponents with cancer severity and poorer prognosis”.

Comment 4: The conclusion can be improved by indicating the translational value of the findings from this systematic review.

Reply: Thank you for your valuable comment. We carefully revised the conclusion section of this manuscript to encompass more information about the translational value of the findings from this study and more broadly of this field of research (page 11, line 392-395).

Reviewer 2 Report

After reading the manuscript, my concerns are as follows:

1. What period of time was searched in various databases? The time limit was mentioned as to the 17th of June 2022, but the beginning of research studies included to this review was not mentioned. Please, add this information to the Materials section.

2. There was no information about the role of desmoglein 1, (DSG1), DSG2, DSG3, desmocollin 2 (DSC2), integrin beta 4 (ITGB4), laminin gamma chain 2 (LAMC2), and collagen type 17 alpha 1 (COL17A1) in the induction of oral cancers. Please, explain why such crucial information was missed.

3. Line 240: Section 3.3.3. In the title of this section, the authors have mentioned: BPA1, BPAG1e, BP230 and dystonin. Why in the text of this section there was no information on the BPAG1e, BP230 and dystonin? The authors have only described BPA1.

4. Line 251: Section 3.3.4. In the title of this section, the authors have mentioned: Col17, BP180 and COL17A1. Why in the text of this section there was no information on the BP180 and COL17A1?

5. Please, explain abbreviations when they are first mentioned in the text. Please, avoid the same acronym to describe various things. For instance (IF) – a) intermediate filament (IF); b) immunofluorescence (IF).

6. Discussion. Please avoid repeating information presented in the results section and discussion. Some parts of the discussion repeat information already presented in the results section. Please, remove the duplicated information.

Author Response

Reviewer 2

After reading the manuscript, my concerns are as follows:

Comment 1: What period of time was searched in various databases? The time limit was mentioned as to the 17th of June 2022, but the beginning of research studies included to this review was not mentioned. Please, add this information to the Materials section.

Reply: Thank you for your valuable comment. We did not apply any restriction in terms of publication year to ensure we were capturing all data available in each database. We have added now this missing information as per your suggestion and the specific sentence on page 2 reads as follows: “Electronic literature searches were conducted on June 17th, 2022, using Scopus, OVID MEDLINE, OVID EMBASE and Web of Science databases with no publication year restrictions”.

Comment 2: There was no information about the role of desmoglein 1, (DSG1), DSG2, DSG3, desmocollin 2 (DSC2), integrin beta 4 (ITGB4), laminin gamma chain 2 (LAMC2), and collagen type 17 alpha 1 (COL17A1) in the induction of oral cancers. Please, explain why such crucial information was missed.

Reply: Thank you for your valuable comment. Our attention to the role of desmosomal components in oral disease already fostered us to publish two interesting review articles a few years ago:

- Celentano A, Mignogna MD, McCullough M, Cirillo N. Pathophysiology of the Desmo-Adhesome. J Cell Physiol. 2017 Mar;232(3):496-505. Review. PMID: 27505028.

- Celentano A, Cirillo N. Desmosomes in disease: a guide for clinicians. Oral Dis. 2016 Jun 22. Review. PMID: 27329525.

However, as you know Type 1 and 2 hemidesmosomes include overall just five main elements: integrin α6β4, plectin in its isoform 1a, tetraspanin protein CD151, BPAG1e, or bullous pemphigoid antigen isoform e, and BPAG2 (also known as BP180 or type 17 collagen). Desmosomal components, such as desmogleins and desmocollins and their role in oral carcinogenesis were beyond the stated scope of this review whose focus and purpose was strictly the role of Hemidesmosomal components in oral carcinogenesis. We clearly indicated this in the title and in the aims of our study. We do not exclude that in the near future we may be able to dedicate a further piece of work to the role of the desmosomal complex in oral carcinogenesis. Conversely, the role of integrin beta 4 (both as integrin α6β4 heterodimer and as single subunit) and collagen type 17 alpha 1 (also known as BPAG2/Col17/BP180/COL17A1) in oral carcinogenesis has been extensively discussed in our study.

Comment 3: Line 240: Section 3.3.3. In the title of this section, the authors have mentioned: BPA1, BPAG1e, BP230 and dystonin. Why in the text of this section there was no information on the BPAG1e, BP230 and dystonin? The authors have only described BPA1.

Reply: Thank you for this comment. The reason is that all those names are just alternative names/synonyms of the same target protein BPA1. Furthermore, all possible alternative  names/synonyms include: 230 kDa bullous pemphigoid antigen; 230/240 kDa bullous pemphigoid antigen; BP230; BP240; BPA; BPA1; BPAG1; Bullous pemphigoid antigen; Bullous pemphigoid antigen 1; bullous pemphigoid antigen 1, 230/240kDa; CATX-15; CATX15; D6S1101; DKFZp564B2416; DMH; DST; DT; DYST; dystonia musculorum protein; dystonin; EBSB2; FLJ46791; Hemidesmosomal plaque protein; HSAN6; KIAA0465; KIAA0728; KIAA1470; MACF2; trabeculin-beta (Ref: https://www.phosphosite.org/proteinAction.action?id=1660800&showAllSites=true)

Comment 4: Line 251: Section 3.3.4. In the title of this section, the authors have mentioned: Col17, BP180 and COL17A1. Why in the text of this section there was no information on the BP180 and COL17A1?

Reply: Thank you for your comment. This issue raised is very similar to the comment n. 3 of this reviewer. All those names are just alternative names/synonyms of the same target protein.

Comment 5: Please, explain abbreviations when they are first mentioned in the text. Please, avoid the same acronym to describe various things. For instance (IF) – a) intermediate filament (IF); b) immunofluorescence (IF).

Reply: Thank you for your valuable comment. We revised all the abbreviations throughout the text to ensure the full spelling at first use. We also acknowledge that the use of two “IF” abbreviations could be confusing, therefore we decided to leave the abbreviation “IF” for the Intermediate filaments, while we re-label the abbreviation for “Immunofluorescence” as “IFL”.

Comment 6: Discussion. Please avoid repeating information presented in the results section and discussion. Some parts of the discussion repeat information already presented in the results section. Please, remove the duplicated information.

Reply: Thank you for your valuable comment. We carefully revised the discussion section to remove redundant content.

Round 2

Reviewer 2 Report

No further concerns to the paper